# LASSO Regression-Based Diagnosis of Acute ST-Segment Elevation Myocardial Infarction (STEMI) on Electrocardiogram (ECG)

**DOI:** 10.3390/jcm11185408

**Published:** 2022-09-15

**Authors:** Lin Wu, Bin Zhou, Dinghui Liu, Linli Wang, Ximei Zhang, Li Xu, Lianxiong Yuan, Hui Zhang, Yesheng Ling, Guangyao Shi, Shiye Ke, Xuemin He, Borui Tian, Yanming Chen, Xiaoxian Qian

**Affiliations:** 1Department of Cardiology, The Third Affiliated Hospital of Sun Yat-sen University, Guangzhou 510630, China; 2Department of Endocrine and Metabolic Diseases, Guangdong Provincial Key Laboratory, The Third Affiliated Hospital of Sun Yat-sen University of Diabetology, No. 600, Tianhe Road, Guangzhou 510630, China; 3Department of Science and Technology, The Third Affiliated Hospital of Sun Yat-sen University, No. 600, Tianhe Road, Guangzhou 510630, China; 4Department of Medical Ultrasound, Guangzhou First People’s Hospital, School of Medicine, South China University of Technology, No. 1, Panfu Road, Guangzhou 510641, China

**Keywords:** ST-segment elevation myocardial infarction, electrocardiogram, logistic least absolute shrinkage and selection operator regression model, left anterior descending artery disease

## Abstract

Electrocardiogram (ECG) is an important tool for the detection of acute ST-segment elevation myocardial infarction (STEMI). However, machine learning (ML) for the diagnosis of STEMI complicated with arrhythmia and infarct-related arteries is still underdeveloped based on real-world data. Therefore, we aimed to develop an ML model using the Least Absolute Shrinkage and Selection Operator (LASSO) to automatically diagnose acute STEMI based on ECG features. A total of 318 patients with STEMI and 502 control subjects were enrolled from Jan 2017 to Jun 2019. Coronary angiography was performed. A total of 180 automatic ECG features of 12-lead ECG were input into the model. The LASSO regression model was trained and validated by the internal training dataset and tested by the internal and external testing datasets. A comparative test was performed between the LASSO regression model and different levels of doctors. To identify the STEMI and non-STEMI, the LASSO model retained 14 variables with AUCs of 0.94 and 0.93 in the internal and external testing datasets, respectively. The performance of LASSO regression was similar to that of experienced cardiologists (AUC: 0.92) but superior (*p* < 0.05) to internal medicine residents, medical interns, and emergency physicians. Furthermore, in terms of identifying left anterior descending (LAD) or non-LAD, LASSO regression achieved AUCs of 0.92 and 0.98 in the internal and external testing datasets, respectively. This LASSO regression model can achieve high accuracy in diagnosing STEMI and LAD vessel disease, thus providing an assisting diagnostic tool based on ECG, which may improve the early diagnosis of STEMI.

## 1. Introduction

ST-segment elevation myocardial infarction (STEMI) is the leading cause of heart failure and death [1]. Early diagnosis of STEMI can effectively shorten the revascularization time, which helps doctors adopt precise treatment strategies, thereby reducing the incidence of heart failure and mortality [2]. Coronary angiography (CAG) is the gold standard for diagnosing STEMI, but it is invasive, time-consuming, and expensive. Electrocardiography (ECG) is a noninvasive and effective screening tool to detect STEMI in patients with chest pain [3]. However, faced with a large number of ECGs, the diagnosis of STEMI has become a great challenge for clinical physicians [4,5].

Although most ST elevation in the ECG indicates myocardial ischemia, there are many nonischemic etiologies to induce ST elevation, such as bundle branch block, ventricular hypertrophy, ventricular preexcitation, premature ventricular beat, and pacemaker rhythm [6]. These changes can mask the STEMI-triggered ST-segment elevation and cause real STEMI to be missed. In addition, the decrease in ECG amplitude can lead to missed diagnoses of STEMI, such as pulmonary disease, effusion, or anasarca [7]. Moreover, the diagnostic accuracy of ECG varies by level of the doctor, especially in primary and community hospitals. Therefore, the rapid and accurate diagnosis of STEMI based on ECG is still an urgent issue that needs to be resolved.

With the rapid growth of machine learning technologies, several successful ECG automatic diagnosis algorithms have achieved positive results for the detection of STEMI patients [8]. There are several machine learning algorithms for analysing ECG, which have solved the problems of noise reduction, feature extraction, detection of arrhythmia, and left ventricular hypertrophy [9,10,11,12]. For instance, an artificial intelligence (AI) network can analyze STEMI ECG through signal transformation and analysis, as well as automated ECG feature extraction [13,14]. However, these models have insurmountable defects, as most of them were trained and validated using data from the MIT-BIH database (PhysioNET) and the PTB database (physiobank) [14,15]. Moreover, some research excluded arrhythmias that may affect QRS morphology and ST-segment changes. Recently, a machine learning model was built based on real-world ECG data to detect ACS, but it failed to confirm the accuracy by comparison to CAG [16]. Due to the above reasons, there were rare machine learning models that can effectively detect STEMI with arrhythmias and diagnose infarct-related arteries in myocardial infarction.

In this study, we established a real-world ECG database, which was confirmed by gold-standard CAG. Moreover, a LASSO regression model was built and trained to diagnose STEMI and determine the location of infarct-related arteries, followed by a comparison of the diagnostic performance between machine learning and doctors.

## 2. Methods

### 2.1. Study Design

We enrolled patients who underwent CAG at the Third Affiliated Hospital of Sun Yat-sen University (Cohort 1) and the Guangzhou First People’s Hospital (Cohort 2) from Jan 2017 to Jun 2019. The inclusion criteria were as follows: older than 18 years, no prior history of myocardial infarction or percutaneous coronary intervention (PCI) or coronary artery bypass graft (GABG), and CAG for any reason. The exclusion criteria were as follows: excessive ECG noise, multiple vascular diseases, no CAG performed during the first 24 h at the onset of symptoms (such as angina pain, chest pain, backache, shoulder pain, and stomach ache), and incomplete baseline data. This study was approved by the Human Ethics Boards of the Third Affiliated Hospital of Sun Yat-sen University and Guangzhou First People’s Hospital.

We designed two stages to classify STEMI and the location of the infarct-related arteries. The first stage (Model 1) was to establish a model to distinguish between control and STEMI patients. The second stage (Model 2) was to establish a model to identify the control, LAD, LCX, and RCA.

For the model development, we randomly allocated the data into training, validation, and testing datasets based on the ratio 3:1:1. The performance of the model was validated by internal and external testing datasets. The flow chart for collecting ECG and constructing LASSO regression was shown in Figure 1.

### 2.2. Study Setting and Data Collection

A standard protocol containing demographics, complications, laboratory tests, 12-lead resting ECG reports, and 180 ECG features, along with responding CAG reports, was used to collect data. Twelve-lead ECG data were collected prior to thrombolysis therapy or PCI therapy and then stored for analysis. A cardiologist committee was composed of two board-certified practicing cardiac electrophysiologists and one board-certified practicing cardiologist. All ECG data were reviewed by the two committee members, who made the main diagnosis of STEMI and arrhythmia. A third committee member reassessed the ECG data when there was discordance between the first two members. Cases without a majority opinion after the cardiologist committee reviews were excluded. Acute STEMI was diagnosed according to clinical manifestations, ECG changes, and myocardial enzyme changes based on the Fourth Universal Definition of Myocardial Infarction [5]. The definitions of ST-elevation at J points are based on the American College of Cardiology/American Heart Association and the European Society of Cardiology STEMI guidelines. ST-elevation is defined by the Fourth Universal Definition of Myocardial Infarction consensus statement: (1) ST elevation in V2-V3 ≥ 2 mm in men ≥ 40 years, ≥2.5 mm in men < 40 years, or ≥1.5 mm in women, or ST-elevation ≥ 1 mm in other leads; (2) ST depression ≥ 0.5 mm; or (3) T-wave inversion ≥ 1 mm in leads with a prominent R wave or R/S ratio ≥ 1. For the cases with left bundle branch block, the Smith-Modified Sgarbossa Criteria was used to define the STEMI [17]. The location of infarct-related arteries was confirmed by CAG.

### 2.3. Ecg Data

All subjects underwent a resting surface ECG by a physician, with the subjects lying in the supine position (paper speed: 25 mm/s, calibration: 1 mv = 10 mm, ECGNET Vision 3.0, SanRui Electronic Technology, Guangdong, China). All ECG data were digital, standard, 10-s, 12-lead ECG. The ECG data of each patient were marked with the study ID. Poor ECG data were excluded by two independent doctors according to the flow chart. The sampling rate of ECG was 1000 Hz. Raw ECG data were stored in The Third Affiliated Hospital of Sun Yat-sen University Clinic cloud database. A total of 180 ECG features were automatically obtained by an ECG management system (ECGNET Vision 3.0, SanRui Electronic Technology, Guangdong, China). The interpretation of ECG features is shown in Figure 2 and Appendix A. The two major components of the features are the distance between each wave and the amplitude of each wave.

### 2.4. Lasso Regression

To avoid overfitting and simplifying the model, LASSO regression was used to automatically screen ECG features and to push the coefficient estimates toward zero (Figure 1C). Furthermore, we tuned the parameter selection in the LASSO model via minimum criteria. The area under the curve (AUC) of the receiver operating characteristic curve was plotted. A coefficient profile plot was produced against the $\log_{\left(\lambda \right)}$ sequence. A vertical line was drawn at the value selected, optimizing (λ). Dotted vertical lines were drawn at the optimal values by using the minimum criteria and 1 standard error of the minimum criteria (the 1-SE criteria). Finally, the remaining variables after multivariable analysis were regarded as potential risk factors and included in the training cohort. The accuracy of the model was evaluated by ROC curve, AUC of the receiver operating characteristic, sensitivity, specificity, positive predictive value (PPV), and negative predictive value (NPV). The accuracy was obtained from the best cutoff point in the ROC curve based on the maximum Youden index.

### 2.5. Comparative Test

The internal testing data of ECG images were stored in JPG format and tested in a comparative test. The diagnosis was performed blindly and independently. Four levels of doctors were included: experienced cardiologists, emergency physicians, internal medicine residents, and medical interns. Each level contained four doctors. Experienced cardiologists referred to those who had been engaged in the cardiovascular field for more than five years. Emergency doctors referred to those who had worked in the emergency department for more than two years. Internal medicine residents were those with medical licenses but who had not majored in cardiology. Medical interns had completed the theoretical study of cardiology and electrocardiography.

### 2.6. Statistical Analysis

Continuous data are shown as the mean value ± standard deviation, and categorical data are displayed as absolute numbers and percentages. Two independent sample *t*-tests in normal distribution were used to analyze variables between two groups. Pearson’s chi-square test was used to analyze categorical data. Values of *p* < 0.05 are indicated as significant. Statistical analysis was performed in R (version 3.6.2: R Core Team, Vienna, Austria).

## 3. Results

### 3.1. Clinical Characteristics

In total, 820 subjects were enrolled in this study, with 475 control individuals and 259 STEMI individuals from the Third Affiliated Hospital of Sun Yat-sen University (Cohort 1, Figure 1A) and 59 STEMI patients and 27 control patients from the Guangzhou First People’s Hospital for external validation (Cohort 2, Figure 1B). Baseline characteristics are shown in Table 1 and Table 2 and Appendix A. Control subjects were younger than STEMI patients (55.1 ± 12.4 vs. 60.6 ± 12.8) years. There was no significant difference in terms of age between Cohort 1 (57.0 ± 13.0) years and Cohort 2 (59.2 ± 10.6) years. Male patients accounted for 44.6% and 18.2% in the control and STEMI groups and 35.1% and 27.9% in Cohort 1 and Cohort 2, respectively. The prevalence rates of diabetes, chronic kidney diseases, and family history between the STEMI and control groups were not significantly different. Although the ratio of STEMI in Cohort 1 and Cohort 2 are different, the proportion of each infarct-related artery is balanced in Cohort 1 and Cohort 2. The interpretation of variables is shown in Appendix A.

### 3.2. Model Performance

The details of 180 raw ECG features between the control and STEMI groups are shown in Appendix A. ECG features between control and STEMI, between Cohort 1 and Cohort 2, locations of infarct-related arteries, and among different data sets are shown in Appendix A. These significant ECG features were subjected to LASSO regression analysis to construct the diagnosis model. To avoid overfitting and the simplicity of the model, the model with less than one standard error and fewer variables was selected for comparison with the minimum error, and the confidence interval of the general model error was narrower. The diagnosis model performed the best when 14 features were included for STEMI screening (Table 3, Figure 3A) and 4 features were included for LAD location (Table 3, Figure 3C). The proportion of STEMI combined above ECG abnormal phenomenon is 10% (82/821). The performance of our model is shown in Appendix A. The calculation of the regression coefficient is visualized in Figure 3B to detect STEMI and in Figure 3D for the location of LAD vessel disease. In short, most of these ECG features were closely represented by the amplitude of the J point, ST-segment, and Q wave.

The model performance is shown in Table 3. After training and model optimization, the AUCs of STEMI were 0.94 (95% CI: 0.90–0.98) in the internal testing dataset and 0.93 (95% CI: 0.88–0.98) in the external testing dataset. The accuracies were 0.85 and 0.84 in the internal and external testing datasets, respectively. In model 2, we established LASSO regression and logistic regression models to distinguish LAD and RCA/LCX. After optimizing the model, the AUCs were 0.92 (95% CI: 0.83–0.99) and 0.98 (95% CI: 0.95–1) in the internal and external testing datasets, respectively. The accuracies between LAD and RCA/LCX of the internal and external testing datasets were 0.84 and 0.95, respectively. The sensitivities of the internal and external testing datasets were 0.88 and 0.97, respectively. The specificities of the internal and external testing datasets were 0.79 and 0.94, respectively.

Furthermore, we identified STEMI in patients with bundle branch block, ventricular hypertrophy, preexictation, pacing, and arrhythmias. The proportion of STEMI combined above ECG abnormal phenomenon is 10% (82/821). The performance of our model is shown in Appendix A.

### 3.3. The Interpretation of ECG Features

The nomogram to estimate STEMI and LAD vessel disease was built using the training dataset and validated on the internal and external datasets using the LASSO model (Table 4, Figure 4). The final diagnosis model was well calibrated. Calibration curves were drawn for the detection of STEMI and localizing LAD vessel disease for visual comparison (Figure 5). Compared with that of the control group, the amplitude of the ST-segment was significantly different at different distances from the J point.

After screening by the LASSO model, the V4 (PB), V6 (Q), III (TE), AVL (TB), II (R), II (TB), V1 (Q), III (ST 80), V5 (TE), R-Q, and QT intervals were of great significance in the diagnosis of STEMI (Table 4, Figure 4A). These features mean the Q waves of Lead V1 and Lead V6, the beginning of the ST-segment of Lead II, Lead III, Lead AVL, and Lead V5, the ST-segment of Lead III, the beginning of the P wave of V4, the amplitude of the R wave in Lead II, the R-Q interval, and the QT interval. These features focused on the ST change, the Q wave, the amplitude of R, and the QT interval. These leads were mainly concentrated in the inferior and left chest leads.

In the second stage, V1 (Q), V2 (Q), V2 (TB), and V3 (ST40) contributed to the diagnostic model of LAD (Table 4, Figure 4B). These features represented the Q wave in Lead V1 and Lead V2, the beginning of the ST-segment of V2 and the ST-segment of V3. These ECG features focused on the change in the ST-segment. The lead position was located in the anterior septal leads.

### 3.4. Comparative Test

In model 1, experienced cardiologists, emergency physicians, internal medicine residents, and medical interns achieved AUCs of 0.92 (0.90–0.95), 0.86 (0.82–0.89), 0.83 (0.80–0.86), and 0.76 (0.72–0.80), respectively, suggesting that the more experienced doctors had higher accuracy in diagnosing STEMI. Our model surpassed all levers of doctors. In addition, cardiologists gained the highest sensitivity (0.85), specificity (0.86), PPV (0.76), and NPV (0.91) (Table 5). To identify the infarct-related arteries in model 2, the trend of performance in different levels of doctors is similar to that in model 1. Compared to doctors, the LASSO model obtained a sensitivity of 0.85 and 0.88 in model 1 and model 2, respectively, which were able to compensate for the weakness of low sensitivity by doctor diagnosis (Table 5).

## 4. Discussion

In this study, we reported a machine learning algorithm based on 12-lead ECG to detect STEMI, which showed high sensitivity and specificity in distinguishing STEMI, with an AUC of 0.94. In addition, we demonstrated that the LASSO model improved the diagnostic accuracy of detecting LAD lesions, with a low false positive rate and a high NPV.

The first finding of this study was that the LASSO method was able to reduce the regression coefficient and cut 180 candidate ECG features down to 14 potential predictors in model 1 and 4 potential predictors in model 2. This method preceded traditional methods of choosing the ECG index according to the strength of the univariable association with outcome.

The innovation of data science, especially machine learning and AI, has brought revolutionary changes to the diagnosis of ECG, breaking through previous diagnosis concepts [18]. Previous ECG signal acquisition, filtering, and processing capabilities were performed by ANN, SVM, AdaBoost, and naive Bayes classifiers, with ACCs reaching 99.7% [19]. These algorithms extracted the signal of the original ECG diagram and detected the peak point of the QRS waveform by adopting a peak-detection algorithm. However, identifying the ST-segment and T wave changes is much more complex than identifying QRS waveforms. To avoid overfitting, random forest can be utilized in practical ECG applications, especially wearable medical devices and implanted medical devices, for wave detection and arrhythmia classification [20,21]. Many neural networks use a convolution process to mimic how the visual cortex addresses images. Unlike many other machine learning methods, deep learning models not only associate input features with outputs of interest but also learn the features from the original data [18]. Recently, a new model, STA-CRNN, has been reported to recognize most arrhythmias, reaching an average F1 score of 0.835. Through visualization, it is proven that the learning characteristics of STA-CRNN are consistent with clinical judgment [22].

AI technology is becoming smarter and more accurate in detecting arrhythmia, but it is still incompetent in the diagnosis of acute myocardial infarction. Yifan Zhao et al. proposed a Res-Net block to differentiate STEMI ECG from control ECG, with an AUC of 0.99, which was similar to that of cardiologists [8]. However, these models cannot identify the infarct-related arteries of STEMI.

The second advantage of this study is that we used real-world ECG data, which were further confirmed by CAG in both the control and STEMI groups. Most previous AI algorithms were based on the MIT-BIH database (PhysioNET) [19] or the PTB database (physiobank) [23], both of which have small sample sizes. For instance, the MIT-BIH Arrhythmia Database consists of 549 records from 290 subjects, including 148 cases of myocardial infarction and 52 healthy controls, containing 48 half-hour excerpts of two-channel ambulatory ECG recordings.

Unlike previous databases, our datasets are superior, as we included abnormal ECG phenomena that affected ST-segment changes, such as complete left bundle branch block, complete right bundle branch block, ventricular pre-excitation, premature ventricular beats, and ventricular tachycardia. In our study, this type of abnormal ECG phenomenon accounted for 9.5% (70/734) in Cohort 1 and 30% (26/86) in Cohort 2, with high proportions of ventricular premature beats, complete right bundle branch block, and left ventricular hypertrophy. Nestelberger et al. found that AMI occurred in approximately 30% of complete left bundle branch blocks. Using the modified Sagarbossa combined with 0/1 h or 0/2 h hs-cTnT could increase the diagnostic rate to above 90% [24]. Although previous studies have suggested that a new complete left bundle branch block should be cautiously extrapolated to AMI, it is still necessary to identify STEMI in patients with left bundle branch block accompanied by chest pain [25]. Ventricular pre-excitation likely manifests as false myocardial infarction with abnormal Q waves and ST-segment elevation or other symptoms that cover up real myocardial infarction and can lead to clinical misdiagnosis and missed diagnosis [26,27]. Patients with left ventricular hypertrophy have a higher incidence of myocardial infarction and stroke [28]. In survivors of myocardial infarction, left ventricular hypertrophy suggests more severe structural and functional damage to the heart [29]. In this study, our algorithm can still achieve good accuracy in a dataset containing several kinds of abnormal ECG phenomena.

Compared with deep learning, the ECG features screened by LASSO regression were more interpretable. V1 (Q) and V6 (Q) suggested pathological Q wave, AVL (TB) and II (TB) suggested J-point elevation, and III (ST80) suggested ST-segment elevation. These abnormal indices compose the diagnostic model of STEMI. Pathological Q wave and ST-segment elevation are important indicators of STEMI [30]. Another new finding of this study was that we identified prolongation of the QT interval and decrease in the R wave peak as important markers of ECG changes in STEMI. Interestingly, we also noticed that V1 (Q), V2 (Q), V2 (TB) and V3 (ST40) contributed to the diagnosis of LAD and were related to LAD innervating the anterior ventricular septum, the left ventricular anterior wall and the right ventricular anterior wall.

There were some limitations of this study. First, our LASSO model can only discriminate the infarct-related arteries between LAD and RCA/LCx. Because the occlusion of LCx or RCA is the major reason for inferior myocardial infarction (AIMI), it is difficult to diagnose the infarct-related arteries that is caused by RCA or LCx occlusion according to 12-lead ECG. There are several ECG criteria to solve this problem, and we will explore a new ML model with knowledge fusion. Second, the sample size of this retrospective study was small, especially the external test dataset. Third, in this study, patients with multiple vessel lesions were excluded, and patients with multiple vessel lesions accounted for more than 40–50% of patients with myocardial infarction [31]. The ECG pattern of STEMI with multiple vessels is variable and atypical. The change in ECG depends on the infarction area and the contribution degree of each vessel. Our model just tries to explore the differential diagnosis of infarct-related arteries in patients with a single vessel disease. Figuring out the infarct-related arteries in patients with multi-vessel coronary artery disease is still a major challenge for clinical physicians. We will explore the diagnostic efficacy of ECG in patients with multiple vessels in the real-world using the LASSO method in further studies. Moreover, further research is needed to clarify the location of the lesion (proximal versus distal) and the size of the infarct-related arteries. In real-world data, the incidence of STEMI-combined ECG abnormal phenomena, such as bundle branch block (left and right) or arrhythmias (such as AF and VT), is low. Because the real-world data are used in our study, the proportion of STEMI combined above ECG abnormal phenomenon is 10% (82/821), and the AUC is 0.879 (0.797–0.961). In order to verify the accuracy and robustness of the algorithm, we plan to construct a prospective study. In the future, we will embed this model into the application system so that clinicians can directly import ECG data and output results.

In this study, we constructed a machine learning model that provided good performance for detecting STEMI based on 12-lead ECG features, which were autoextracted from a real-world database. This machine learning model performed exceptionally with high diagnostic accuracy similar to that of experienced cardiologists, especially in the location of LAD vessel disease.

## Figures and Tables

**Figure 1 jcm-11-05408-f001:**
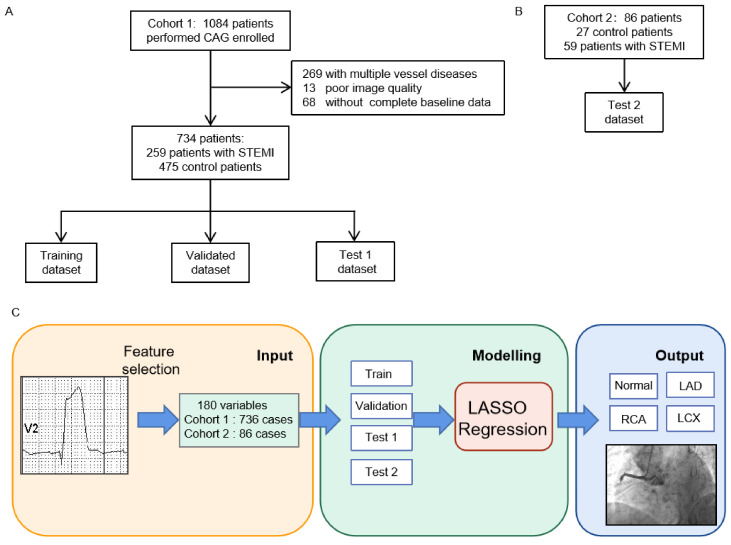
Flow chart for collecting ECG data and constructing LASSO regression. (**A**) The collection steps of Cohort 1 for the training, validation, and internal testing datasets. (**B**) The selection steps of the external testing dataset in Cohort 2. (**C**) The construction of the LASSO model. STEMI = ST-segment elevation myocardial infarction.

**Figure 2 jcm-11-05408-f002:**
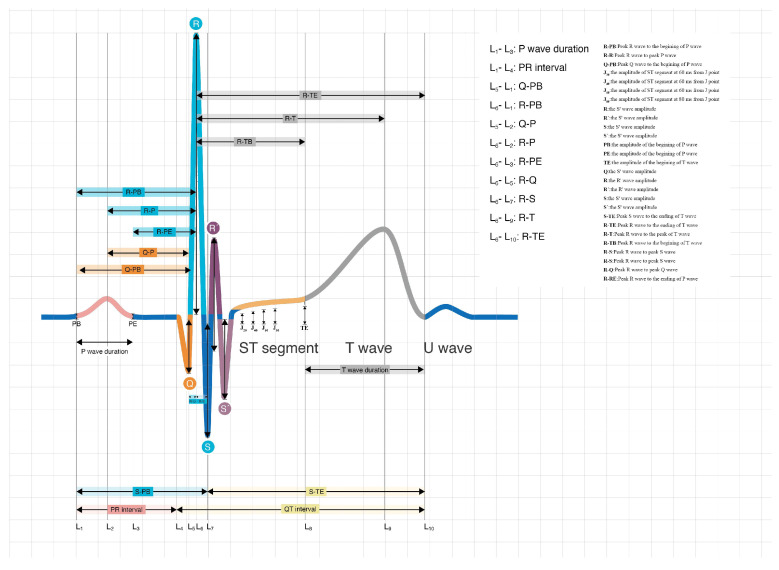
Automatic feature extraction from ECG. The abbreviations of ECG features are shown in Appendix A.

**Figure 3 jcm-11-05408-f003:**
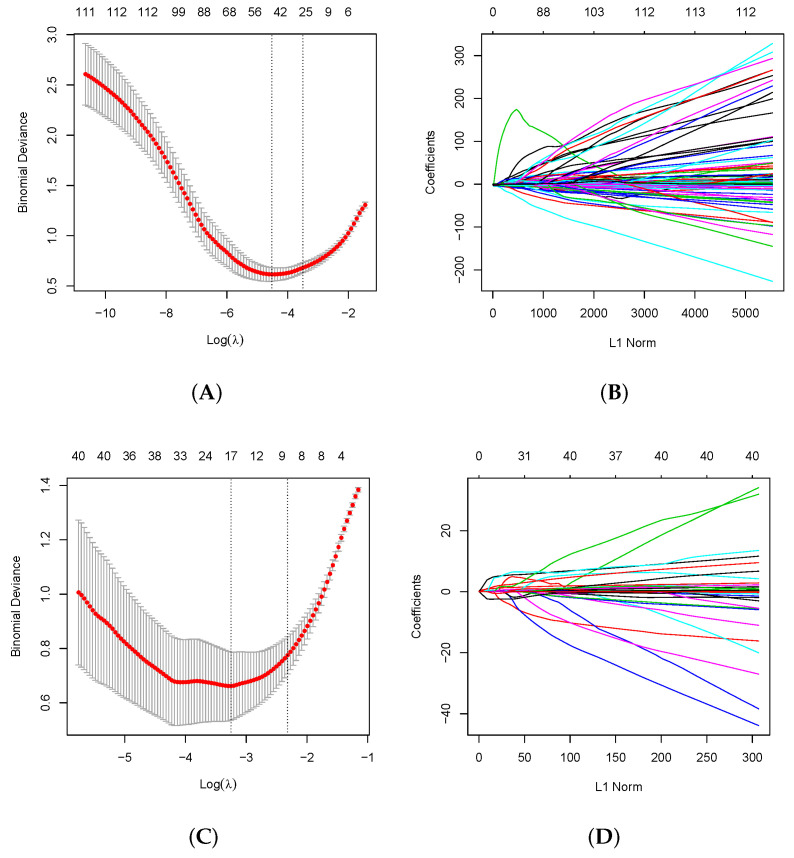
Feature selection by LASSO. (**A**) To differentiate STEMI ECG from control ECG, LASSO regression was used for variable screening. The results showed that 40 variables were retained when the error was the smallest; that is, the place corresponding to the dotted line on the left. To avoid overfitting and simplicity of the model, no more than one standard error was selected compared with the minimum error, and 14 variables were retained, which corresponded to the place on the dotted line on the right. (**B**) LASSO coefficient profiles of the 180 ECG features. A coefficient profile plot was produced against the $\log_{\left(\lambda \right)}$ sequence. A vertical line was drawn at the selected optimizing value (λ), which resulted in 14 nonzero coefficients. (**C**) To discriminate LAD and RCA/LCX, the results showed that when the error was minimal, approximately four variables were reserved; that is, the place corresponding to the dotted line on the left. (**D**) The LASSO screening results were further stepwise screened based on AIC, and four variables were retained.

**Figure 4 jcm-11-05408-f004:**
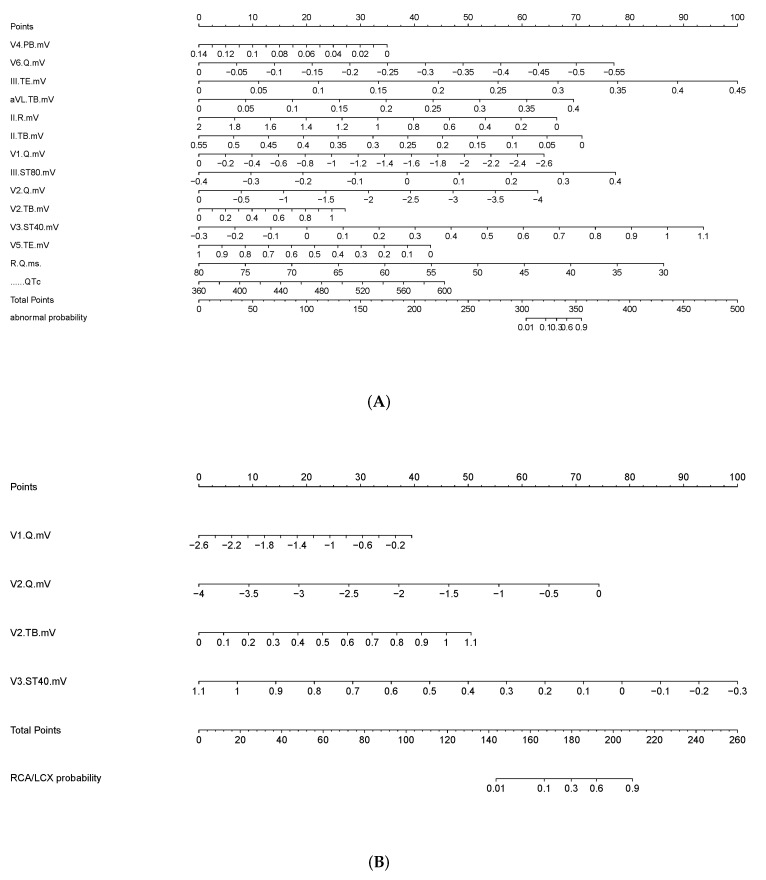
Nomogram for the prediction of ECG features for STEMI and LAD vessel disease. The magnitude of risk prediction can be quantified by drawing a vertical line connecting the value of each variable with the point score at the top of the nomogram (the “point” line). The individual scores are added to generate a total score, which is drawn along the “total score” line and the “risk” line at the bottom of the nomogram. (**A**) Nomography for the diagnosis STEMI. (**B**) Nomography for the location of LAD vessel disease. V4(PB): the amplitude of the beginning of P wave in Lead V4; V6(Q): the Q wave amplitude in Lead V6; III(TE): the end of T wave in Lead III; AVL(TB): the beginning of T wave in Lead AVL; II(R): the amplitude of R wave in Lead II; II(TB): the amplitude of the beginning of T wave in Lead II; V1(Q): the Q wave amplitude in Lead V1; III(ST80): the amplitude of ST-segment at 80 ms from J point in Lead III; V2(Q): the Q wave amplitude in Lead V2; V2(TB): the beginning of T wave in Lead V2; V3(ST40): the amplitude of ST-segment at 40 ms from J point in Lead V3; V5(TE): the amplitude of the end of T wave in Lead V5; R-Q interval: the interval between Peak R wave and peak Q wave; QTc interval: the interval between the beginning of Q wave and the end of T wave.

**Figure 5 jcm-11-05408-f005:**
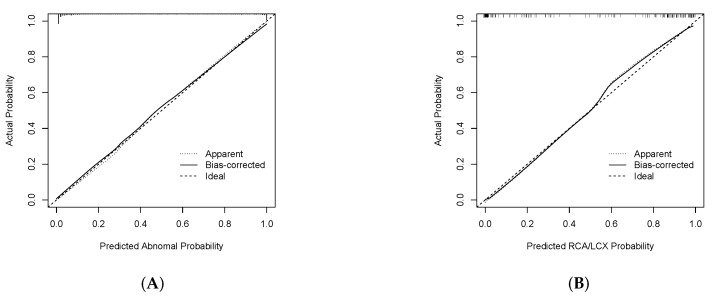
Calibration curve for (**A**) detecting STEMI and for (**B**) LAD vessel disease on the basis of the LASSO regression analysis.

**Table 1 jcm-11-05408-t001:** Baseline characteristics in Cohort 1 and Cohort 2.

	Cohort 1	Cohort 2	*p*
n	734	86	
Age (years)	57.0 ± 13.0	59.2 ± 10.6	0.080
Sex (female)	258 (35.1%)	24 (27.9%)	0.030
Diabetes mellitus	147 (20.0%)	9(10.5%)	0.033
Hypertension	321 (43.7%)	16(18.6%)	0.000
Chronic kidney disease	15 (2.0%)	4(4.7%)	0.129
CVD family history	38 (5.2%)	5(5.8%)	0.797
BUN (mmol/L)	5.17 ± 2.6	7.13 ± 3.96	0.000
Cr (mmol/L)	78.31 ± 53.68	91.93 ± 42.92	0.024
CHOL (mmol/L)	4.55 ± 1.24	4.27 ± 1.33	0.047
TG (mmol/L)	1.63 ± 1.1	1.42 ± 1.15	0.086
HDL-C (mmol/L)	1.09 ± 0.34	1.08 ± 0.35	0.909
LDL-C (mmol/L)	2.85 ± 1.03	2.65 ± 1.11	0.086
CK-MB (U/L)	17.13 ± 40.49	11.84 ± 13.05	0.010
STEMI	259 (35.2%)	59 (68.6%)	0.022
Localization of infarct-related arteries, n			
LAD	128 (49.4%)	30 (50.8%)	0.912
RCA	95 (36.6%)	20 (33.9%)	0.862
LCX	36 (13.9%)	9 (15.3%)	0.831
ECG abnormal phenomenon			
Complete left bundle branch block	2 (0.3%)	1 (1.2%)	0.034
Ventricular premature beat	28 (3.8%)	7 (7.8%)	0.077
Preexcitation syndrome	3 (0.4%)	1 (1.2%)	0.010
Complete right bundle branch block	15 (2.0%)	6 (7.0%)	0.008
Left ventricular hypertrophy	16 (2.2%)	10 (11.6%)	0.000
Atrial fibrillation	5 (0.7%)	1 (1.2%)	0.501
Pacing	1 (0.1%)	0 (0)	1.000

CVD: Cardiovascular disease; BUN: Blood urea nitrogen; Cr: Serum creatinine; CHOL: Cholesterol; TG: Triglyceride; HDL-C: High-density lipoprotein cholesterol; LDL-C: Low-density lipoprotein cholesterol; CK-MB: Creatinekinase isoenzyme; LAD: left anterior descending coronary artery; RCA: right coronary artery; LCX: left circumflex coronary artery.

**Table 2 jcm-11-05408-t002:** Baseline characteristics among different datasets.

	Training Dataset	Validation Dataset	Internal Testing Dataset	External Testing Dataset	*p*
n	445	144	145	86	
Age (years)	56.8 ± 13.5	57.8 ± 12.3	57.1 ± 12.5	59.2 ± 10.6	0.409
Sex (female)	148 (33.3%)	51 (35.4%)	59 (40.7%)	24 (27.9%)	0.215
Diabetes	86 (19.3%)	23 (16.0%)	38 (26.2%)	9 (10.5%)	0.020
Hypertension	184 (54.3%)	68 (47.2%)	69 (47.6%)	16 (18.6%)	0.000
CKD	7 (1.6%)	3 (2.1%)	5 (3.4%)	4 (4.7%)	0.260
CVD family history	21 (4.7%)	10 (6.9%)	7 (4.8%)	5 (5.8%)	0.758
WBC (×10${}^{9}$/L)	8.0 ± 3.3	8.2 ± 3.0	8.3 ± 3.3	7.2 ± 2.4	0.070
RBC (×10${}^{9}$/L)	4.4 ± 0.9	4.2 ± 0.8	4.3 ± 1.0	4.3 ± 0.8	0.376
HGB (g/mL)	130.6 ± 17.1	128.1 ± 17.8	128.4 ± 18.5	127.0 ± 22.2	0.197
PLT (×10${}^{9}$/L)	210.8 ± 60.5	206.9 ± 58.5	205.9 ± 54.7	227.3 ± 66.2	0.045
ALB (g/L)	39.2 ± 4.2	38.9± 5.0	38.8 ± 4.5	39.5 ± 4.9	0.676
GLB (g/L)	24.8 ± 4.2	24.3 ± 4.2	24.5 ± 4.6	26.6 ± 4.7	0.001
K (mmol/L)	3.59 ± 0.54	3.6 ± 0.56	3.62 ± 0.54	4.2 ± 0.65	0.000
Na (mmol/L)	137.44 ± 4.4	136.8 ± 4.04	136.58 ± 5.03	141.7± 3.07	0.000
Ca (mmol/L)	1.98 ± 1.05	1.87 ± 0.34	1.94 ± 0.32	2.06 ± 0.24	0.312
GLU (mmol/L)	6.84 ± 3.07	7.34 ± 3.45	6.86 ± 2.92	6.77 ± 1.9	0.335
BUN (mmol/L)	5.14 ± 2.25	5.01 ± 2.93	5.43 ± 3.21	7.13 ± 3.96	0.000
Cr (mmol/L)	75.3 ± 45.3	83.8 ± 75.9	82.1 ± 50.7	91.9 ± 42.9	0.031
CHO L (mmol/L)	4.53 ± 1.19	4.6 ± 1.39	4.54 ± 1.23	4.27± 1.33	0.234
TG (mmol/L)	1.62 ± 1.08	1.69 ± 1.28	1.62 ± 0.94	1.42 ± 1.15	0.325
HDL-C (mmol/L)	1.07 ± 0.33	1.11 ± 0.38	1.11 ± 0.34	1.08 ± 0.35	0.460
LDL-C (mmol/L)	2.84 ± 0.99	2.92 ± 1.14	2.83 ± 1.05	2.65 ± 1.11	0.291
CK-MB (U/L)	16.89 ± 36.82	18.51 ± 51.09	16.51 ± 39.65	11.84 ± 13.05	0.640

CKD: Chronic kidney disease; CVD: Cardiovascular disease; WBC: White blood cell; RBC: Red blood cell; HGB: Hemoglobin; PLT: platelet count; ALB: Seralbumin; GLB: Globulin; K: Potassium; Na: Sodium; Ca: calcium; GLU: Fasting glucose; BUN: Blood urea nitrogen; Cr: Serum creatinine; CHOL: Cholesterol; TG: Triglyceride; HDL-C: High-density lipoprotein cholesterol; LDL-C: Low-density lipoprotein cholesterol; CK-MB: Creatinekinase isoenzyme.

**Table 3 jcm-11-05408-t003:** Diagnostic performance of LASSO in different datasets.

		AUC	Accuracy	SEN	SPE	PPV	NPV
Model 1							
	Train dataset	0.98 (0.97–0.99)	0.93	0.91	0.96	0.92	0.95
	Validation dataset	0.97 (0.94–0.99)	0.89	0.86	0.92	0.85	0.92
	Internal dataset	0.94 (0.90–0.98)	0.85	0.85	0.86	0.76	0.91
	External dataset	0.93 (0.88–0.98)	0.84	0.85	0.82	0.89	0.78
Model 2							
	Train dataset	0.94 (0.91–0.98)	0.92	0.95	0.84	0.86	0.94
	Validation dataset	0.94 (0.85–1.00)	0.88	0.96	0.8	0.83	0.95
	Internal dataset	0.92 (0.83–0.99)	0.84	0.88	0.79	0.81	0.80
	External dataset	0.98 (0.95–1.00)	0.95	0.97	0.93	0.93	0.97

SEN: sensitivity; SPE: specificity; PPV: positive predictive value; NPV: negative predictive value; AUC: area under the curve of receiver operating characteristic.

**Table 4 jcm-11-05408-t004:** Odds ratios of ECG features estimated from the LASSO model.

	Feature	β	se	z	*p*
Mode1					
	(Intercept)	0.714	3.456	0.207	0.836
	V4(PB) (mV)	−33.047	9.296	−3.555	<0.001
	V6(Q) (mV)	−18.559	5.335	−3.479	0.001
	III(TE) (mV)	29.427	5.941	4.953	<0.001
	AVL(TB) (mV)	23.027	4.46	5.163	<0.001
	II(R) (mV)	−4.4	0.835	−5.266	<0.001
	II(TB) (mV)	−17.124	4.374	−3.915	<0.001
	V1(Q) (mV)	−3.264	0.749	−4.358	<0.001
	III(ST80) (mV)	12.8	4.369	2.93	0.003
	V2(Q) (mV)	−2.083	0.768	−2.714	0.007
	V2(TB) (mV)	3.271	1.162	2.816	0.005
	V3(ST40) (mV)	8.861	2.719	3.259	0.001
	V5(TE) (mV)	−5.696	2.265	−2.515	0.012
	R-Q interval (ms)	−0.229	0.059	−3.858	<0.001
	QTc interval (ms)	0.025	0.008	3.33	0.001
Model 2					
	(Intercept)	0.454	0.518	0.878	0.38
	V1(Q) (mV)	1.569	0.691	2.27	0.023
	V2(Q) (mV)	1.917	0.73	2.624	0.009
	V2(TB) (mV)	4.742	1.342	3.535	<0.001
	V3(ST40) (mV)	−7.373	2.274	−3.241	0.001

V4(PB): the amplitude of the beginning of P wave in Lead V4; V6(Q): the Q wave amplitude in Lead V6; III(TE):the end of T wave in Lead III; AVL(TB): the beginning of T wave in Lead AVL; II(R): the amplitude of R wave in Lead II; II(TB): the amplitude of the beginning of T wave in Lead II; V1(Q): the Q wave amplitude in Lead V1; III(ST80): the amplitude of ST-segment at 80 ms from J point in Lead III; V2(Q): the Q wave amplitude in Lead V2; V2(TB): the beginning of T wave in Lead V2; V3(ST40): the amplitude of ST-segment at 40 ms from J point in Lead V3; V5(TE): the amplitude of the end of T wave in Lead V5; R-Q interval: the interval between Peak R wave and peak Q wave; QTc interval: the interval between the beginning of Q wave and the end of T wave.

**Table 5 jcm-11-05408-t005:** Diagnostic performance of different levels of doctors on the comparative test.

		AUC	SEN	SPE	PPV	NPV
Model 1						
	Experienced cardiologists	0.92 (0.90–0.95)	0.88	0.92	0.96	0.97
	Emergency physicians	0.86 (0.82–0.89)	0.74	0.84	0.95	0.97
	Internal medicine residents	0.83 (0.80–0.86)	0.69	0.82	0.96	0.98
	Medical interns	0.76 (0.72–0.80)	0.63	0.77	0.8	0.89
	Our model	0.94 (0.90–0.98)	0.85	0.86	0.76	0.91
Model 2						
	Experienced cardiologists	0.83 (0.78–0.89)	0.72	0.93	0.75	0.94
	Emergency physicians	0.81 (0.76–0.87)	0.70	0.93	0.71	0.93
	Internal medicine residents	0.79 (0.74–0.85)	0.69	0.92	0.63	0.90
	Medical interns	0.68 (0.62–0.73)	0.53	0.87	0.43	0.82
	Our model	0.98 (0.95–1.00)	0.88	0.79	0.81	0.80

SEN: sensitivity; SPE: specificity; PPV: positive predictive value; NPV: negative predictive value; AUC: area under the curve of receiver operating characteristic.

## Data Availability

The custom codes for the diagnosis and discrimination STEMI by LASSO regression are available.

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
