# Peer review of "LASSO Regression-Based Diagnosis of Acute ST-Segment Elevation Myocardial Infarction (STEMI) on Electrocardiogram (ECG)"

_jcm, 2022, doi:10.3390/jcm11185408_

Round 1

Reviewer 1 Report

The article is interesting, valuable and well-written. Its authors should be congratulated.

I have the following comments:

1.     The differentiation only between LAD and RCA/Cx as the culprit vessel is major limitation of the algorithm and should be discussed in the manuscript. What about other culprit vessels (e.g. diagonal artery)? How were they clissfied? Please provide in the text also some data on infarct-related arteries.

2.     The authors state in the introduction section that previous algorithms have problems with identifying STEMI in patients with bundle branch block, ventricular hypertrophy, preexitation, pacing, arrhythmias. In fact, few such patients were included in the present study. Please discuss this fact in the study limitations section.

3.     Line 33: “Aetiologies”. Please change for etiologies as the rest of the manuscript is written in American English.

4.     Line 45: Citations should be listed in the order required by the journal (i.e. [15,18,23,25]), similarly Line 47 should be [1],[6]

5.     Lines 145 and 146: Please add unit (years).

6.     Figures: All abbreviations should be explained after the legend.

7.     Format of references should be modified in order to adhere the journal style.

Author Response

1.

We thank the reviewer for the kindly positive comments. This question is very critical and we have addressed it in the revised Discussion (page 13, line 278-283). Unfortunately, our LASSO model based on 12-lead ECG can only discriminate the culprit vessels between LAD and RCA/LCX. The reasons are listed below.

First of all, due to the technical limitations of the body surface, ECG can’t fully diagnose the secondary branches of the coronary artery. Currently, the diagnosis of secondary branches relies on coronary angiography, such as obtuse marginal artery and first diagonal branch. Moreover, the incident of STEMI caused by secondary branch obstruction is lower than that by main branches, such as LAD, RCA and LCX. The clinical outcome and prognosis cause by secondary branches are also less severe than that of the obstruction of main branches. Therefore, culprit vessels involving the main branches were the major target in this study. It is reported that RCA lesion is mainly related to ST segment elevation in Lead III, and LCX lesion is closely involved in ST segment elevation in Lead II. Due to the complexity, the accuracy of experienced cardiologists in differentiating RCA or LCX based on ECG is still unsatisfied.

In addition, the distribution of coronary arteries in our study was judged by more than 2 cardiologists (who had performed coronary angiography for more than 5 years) according to the SYNTAX score[1]. The data about infarct-related arteries was provided in Page 8 Table 1.

The coronary tree is divided into 16 segments according to the AHA classification. Each segment is given a score of 1 or 2 based on the presence of diseases, and this score is then weighted based on a chart, with values ranging from 3.5 for the proximal left anterior descending artery to 5.0 for left main, and 0.5 for smaller branches.

2.

Thank you for pointing out our limitation. We agree with the reviewer that the real-world data used in our study did not include large number of such patients, as the incident of bundle branch block, ventricular hypertrophy, preexitation, pacing, arrhythmias. in the STEMI is low. We have addressed it in the revised Discussion (page 13, line 295-298).

3.

Response: We apologize for the improper usage of “Aetiiologies” in the original manuscript, which has been corrected to “etiologies” in the revised manuscript. 

4.

Response: We apologize for the citations problems in the original manuscript. The citation order has been checked and corrected throughout the manuscript carefully.

5.

Response: Thank you for pointing out our negligence. We have added unit (year) to line 146 and line 147.

6.

Response: We apologize for the problem. We have added a list of abbreviations used in this manuscript in page 13 Table 6 Abbreviations. 

7.

Response: We apologize for the improper reference styles. We have corrected and changed the reference style according to the journal’s instruction.

[1]Sianos G, Morel MA, Kappetein AP, Morice MC, Colombo A, Dawkins K, van den Brand M, Van Dyck N, Russell ME, Mohr FW, Serruys PW.  EuroIntervention. 2005;1(2):219-27.

Reviewer 2 Report

In their article Wu et al. presented their results regarding the application of machine learning with LASSO logistic regression model in STEMI diagnosis. Noteworthy, they focused on ECG complicated with arrhythmia and identification of culprit vessel.

It has been shown that this approach had high diagnostic value. This article provides interesting results and conclusions, however, I have several comments:

- please organize properly the order of citations

- please use standarized and well-known abbreviations

- the Introduction needs extensive English editing

Author Response

1.

Response: We apologize for citation order problem. We have corrected the citation order in the revised manuscript.

2.

Response: We apologize for the improper abbreviations. We have proofread all the abbreviations and corrected to standardized ones.

3.

Response: We apologize for the language problem. The language presentation has been improved with assistance from a native English speaker with appropriate research background.

Reviewer 3 Report

Lin Wu et al performed a LASSO regression analysis to automatically identify patients with STEMI from digital ECGs. They developed a very robust algorithm with a high AUC. Furthermore, they developed another model that successfully identified the culprit lesion (LAD or non-LAD).

The manuscript is very well-written and the results are impressive. Statistics seem adequate. The conclusions are backed up by the results and citations are adequate. However, the authors should further evaluate the robustness of the algorithm, because in patients with pre-existent ECG abnormalities are a major challenge for both clinicians and algorithms. I therefore invite the authors to include the proportion of ECGs with bundle branch block (left and right), arrhythmias (such as AF and VT), as well as the AUC in patients with those arrhythmias (especially LBBB). How was STEMI diagnosed in those patients? Did the authors use Sgarbossa criteria?

I have the following minor comments:

-        Endpoints (STEMI vs. non-STEMI and LAD vs. non-LAD) should be written in the abstract, as well as specificity and sensitivity for both models.

-        Methods: How was “excessive ECG noise” defined?

-        Why were patients with multiple vessel disease excluded? The culprit lesion may still be identifiable in this patient population.

-        Was the STEMI diagnosis also based on coronary angiography? Were there patients with formal STEMI diagnosis but without culprit lesion or without any coronary lesion in coronary angiography?

Author Response

1.

We appreciate the reviewer’s kind comments. In our study, in order to assess the robustness of the algorithm, we compared the performance of our model in the internal and external testing datasets. Furthermore, we compared the model and different levels of doctors. Our data shown that the performance of our model was consistent in the internal and external testing datasets, and similar to that of experienced cardiologists.

 I very much agree with the reviewers. In fact, patients with STEMI combined pre-existent ECG abnormalities are a big challenge for clinicians and algorithms. In a clinical scenario, to identify pre-existent ECG abnormalities, we should have the previous ECG data of the patients. But in the emergency room, the patient is in an emergency condition, and usually forget to carry previous examination results. Therefore, it is really difficult to find out whether the ECG abnormalities are pre-existent or newly onset by clinicians or our algorithm.

In real-world data, the incidence of STEMI combined ECG abnormal phenomenon, such as bundle branch block (left and right), arrhythmias (such as AF and VT) is low. The proportion of STEMI combined above ECG abnormal phenomenon were reported as follow:

Right Bundle Branch Block (RBBB) has been reported in 5-11% of the acute myocardial infarctions (AMI)[1]. 1.5% LBBB-morphology was seen in STEMI patients[2]. 24% STEMI patients had left ventricular hypertrophy[3]. The proportion of STEMI combined with preexitation is still not clear.

We screened our database for the STEMI combined left or right bundle branch block, arrhythmias (such as AF and VT), ventricular preexcitation from from Jan 2017 to Jun 2019. It is difficult to include more cases with STEMI and above ECG phenomenon. In order to verify the accuracy and robustness of the algorithm, we plan to construct a prospective study cohort.

We have addressed it in the revised Discussion(page 13, line 295-301).

In real-world data, the incidence of STEMI combined ECG abnormal phenomenon, such as bundle branch block (left and right), arrhythmias (such as AF and VT) is low. Because the real-world data are used in our study, the proportion of STEMI combined above ECG abnormal phenomenon is 10%(82/821), and the AUC is 0.879(0.797-0.961). In order to verify the accuracy and robustness of the algorithm, we plan to construct a prospective study.

[1]Figueroa-Triana JF, Mora-Pabón G, Quitian-Moreno J, Álvarez-Gaviria M, Idrovo C, Cabrera JS, Peñuela JAR, Caballero Y, Naranjo M.Acute myocardial infarction with right bundle branch block at presentation: Prevalence and mortality. J Electrocardio. 2021;66:38-42.

[2]Alkindi F, El-Menyar A, Al-Suwaidi J, Patel A, Gehani AA, Singh R, Albinali H, Arabi A.Left Bundle Branch Block in Acute Cardiac Events: Insights From a 23-Year Registry.Angiology. 2015;66(9):811-7.  

[3]Nepper-Christensen L, Lønborg J, Ahtarovski KA, Høfsten DE, Kyhl K, Ghotbi AA, Schoos MM, Göransson C, Bertelsen L, Køber L, Helqvist S, Pedersen F, Saünamaki K, Jørgensen E, Kelbæk H, Holmvang L, Vejlstrup N, Engstrøm T.Left Ventricular Hypertrophy Is Associated With Increased Infarct Size and Decreased Myocardial Salvage in Patients With ST-Segment Elevation Myocardial Infarction Undergoing Primary Percutaneous Coronary Intervention. J Am Heart Assoc . 2017 Jan;6(1):e004823.

2.

For the cases with left bundle branch block, the Smith-Modified Sgarbossa Criteria was used to definite the STEMI[1]. We will supplement the Smith-Modified Sgarbossa Criteria in this Page 3 Line 94-96.

[1]Smith SW et al. Diagnosis of ST Elevation Myocardial Infarction in the Presence of Left Bundle Branch Block using the ST Elevation to S-Wave Ratio in a Modified Sgarbossa Rule. Annals of Emergency Medicine 2012;60:766-776.

3.

Response: Thank you for your suggestion. We have rewrote the abstract as follow:

 To identify the STEMI and non-STEMI, the LASSO model retained 14 variables with AUCs of 0.94 and 0.93 in the internal and external testing datasets, respectively. The performance of LASSO regression was similar to that of experienced cardiologists (AUC: 0.92) but superior (p< 0.05) to internal medicine residents, medical interns, and emergency physicians. Furthermore, in terms of detecting left anterior descending (LAD) or non-LAD, LASSO regression achieved sensitivities of 0.88 and 0.95 in the internal and external testing datasets, respectively.

4.

Response: Thank you for you suggestion .

The excessive noise from sensor circuits that is known as power line interference (PLI). Other noises such as baseline wander (BW) and electrode motion artefact are caused by body motion and poor electrode attachment. Some of the noise sources are uncontrollable—there include body motion, eyelid movement and device circuit noise.

5.

Response: Thank you for pointing this out. We rewrote this content in the limitation(Page 13-14 Line 282-292).

Third, in this study, patients with multiple vessel lesions were excluded, which accounted for more than 40%-50% of patients with myocardial infarction. The ECG pattern of STEMI with multiple vessels is variable and atypical. The change of ECG dependents on the infarction area, the contribution degree of each vessels. Our model just try to explore the differential diagnosis of IRA in patients with single vessel disease. To finger out the IRA in patients with multi-vessel coronary artery disease is still a major challenge for clinical physician. We will explore the diagnostic efficacy of ECG of patients with multiple vessel lesions in the real world by LASSO method in further studies.

6.

Response: Thank you for your questions.Our data are from the database of coronary angiography, thus all patients underwent coronary angiography. The diagnosis of STEMI patients is mainly based on Fourth Universal Definition of Myocardial Infarction (2018)[1]. Strictly speaking, our STEMI patients belong to type 1 STEIM. Therefore, all our STEMI patients have culprit vessels and culprit sites.

 [1]Thygesen K, Alpert JS, Jaffe AS, et al. Fourth Universal Definition of Myocardial Infarction (2018). J Am Coll Cardiol 2018; 72:2231.

Round 2

Reviewer 2 Report

Thank you for your response. The current version is significantly improved.